# Does willpower mindset really moderate the ego-depletion effect? A preregistered replication of Job, Dweck, and Walton (2010)

**Nicholas P. Carruth**[1], **Jairo A. Ramos**[2], **Akira Miyake**[2]*

**1** Department of Psychology, DePaul University, Chicago, IL, United States of America, **2** Department of Psychology and Neuroscience, University of Colorado Boulder, Boulder, CO, United States of America

* akira.miyake@colorado.edu

**Data Availability Statement:** All data files are available from the Open Science Framework database at: https://osf.io/37tgm/.

## Abstract

This article reports a preregistered study in which we attempted to replicate the results of an influential study on the ego-depletion effect reported by Job, Dweck, and Walton in 2010. The original Job et al. study (Study 1, $N = 60$) provided evidence that the ego-depletion effect—a performance decrease on a self-control task after performing another self-control task—occurs only for individuals who hold a belief that their willpower is limited. This moderation of the ego-depletion effect by one's willpower mindset (limited vs. nonlimited) has been interpreted as evidence against a prevalent limited-resource account of self-control. Although this alternative account of the ego-depletion effect has become well-known, the statistical evidence of the original study was on shaky ground. We therefore conducted a preregistered replication of the original study with some methodological improvements. As in the original study, participants ($N = 187$) performed a self-control task (Stroop color-word interference task) after performing the control or depletion version of a letter cancelation task. Despite extensive analyses, we failed to replicate the original results: There was neither a significant main effect of ego depletion nor a significant moderation of this ego-depletion effect by individual differences in willpower mindset. Together with other recent failures to replicate the original moderation effect, our results cast doubts on the claim that an individual's view of whether willpower is limited or not affects one's susceptibility to the ego-depletion effect.

## Introduction

This article reports a preregistered study of an influential finding about the ego-depletion effect first reported by Job et al. [1]. In their original study (Study 1), a total of 60 participants first completed a questionnaire assessing their implicit theories or beliefs about whether willpower is limited or not (hereafter, *willpower mindset*, in analogy to Dweck's [2] influential concept of fixed vs. growth mindset for intelligence). Participants were then randomly assigned to the depletion or control condition and performed the respective versions of a letter cancelation (*e*-crossing) task. As the outcome task, participants performed a manual-response version of

**Funding:** The author(s) received no specific funding for this work.

**Competing interests:** The authors have declared that no competing interests exist.

the color-word Stroop task. Job et al. [1] reported two key findings: (a) a significant overall ego-depletion effect (more Stroop errors in the depletion condition than in the control condition) and (b) a significant moderation of this effect by willpower mindset (only individuals with a limited-willpower mindset demonstrated depletion).

This study is worthy of a rigorous replication for several important reasons. To begin with, the last several years have witnessed major challenges to the ego-depletion effect on its conceptual and empirical grounds [3, 4], especially regarding its replicability. Although Hagger et al.'s [5] early meta-analysis of the ego-depletion effect, based only on published studies, yielded moderate effect sizes (Cohen's $d$ = .62), more recent meta-analyses suggested that, due to the well-known phenomenon of publication bias and possible questionable research practices such as $p$-hacking [6], the actual effect size might be much smaller [7–9] or even nonexistent [10]. Moreover, recent preregistered multilab replication attempts produced nonsignificant results ($d$ = .04 [11] and $d$ = .06 [12]) or a significant yet weak effect size ($d$ = .10 [13]) for the ego-depletion effect (see also Vadillo et al.'s [9] Many Labs 3 results, which also failed to observe a significant ego-depletion effect).

Given that Job et al.'s [1] account must assume that the ego-depletion effect can be reliably observed at least among some individuals (i.e., those holding a limited-willpower mindset), such replicability challenges for the ego-depletion effect itself make it important to replicate the proposed moderating influence of one's willpower mindset on the ego-depletion effect. This is especially true, considering that it has been known that detecting a statistical moderation effect is difficult when it involves individual-differences variables [14].

Second, from a more theoretical standpoint, Job et al.'s [1] original finding is worth replicating because this highly cited work (1,272 citations on Google Scholar Citations on May 26, 2023) has been portrayed as providing a conclusive refutation of a dominant account of the ego-depletion effect, the strength model of self-control proposed by Baumeister and his colleagues [15]. This strength model postulates that self-control relies upon a finite domain-general resource and that exerting self-control depletes this limited resource supply, thereby impairing subsequent self-control performance. Job et al.'s [1] results suggest that one's willpower mindset, not the limited supply of self-control resources per se, may be the more critical variable underlying the ego-depletion effect, a claim that has even permeated the popular media [16–18]. Most notably, in his recent blog entitled "The last thing you need to know about ego depletion," Berkman [16] called Job et al.'s [1] original finding "the biggest bombshell" against the resource-based strength model of self-control: "The ultimate power of a person's beliefs over their willpower completely undermines the premise that willpower draws [upon] an inherently limited resource."

Job et al.'s [1] willpower-mindset account of ego depletion also provided a basis for their subsequent studies. For example, Job's group [19] demonstrated that consuming glucose (but not placebo) drinks reduces the negative impact of ego depletion only among individuals holding a limited-willpower mindset. Moreover, Savani and Job [20] even proposed that one's willpower mindset is so powerful that Indian participants, who as a group tend to believe that exerting self-control is energizing, rather than depleting, exhibit a reverse ego-depletion effect (i.e., exerting self-control *improves*, rather than impairs, subsequent self-control performance).

Despite such major theoretical impacts of Job et al.'s study [1], a close reading of the original article raises serious concerns regarding the robustness and replicability of their novel discovery. To begin with, the sample size was small (total $N$ = 60) for a study that involved a between-subjects factor (depletion vs. control) and sought to detect a statistical (non-crossover) moderation effect involving an individual-differences variable [14].

Moreover, Job et al. [1] conducted their statistical analyses in a highly unconventional manner. Although they conducted hierarchical linear modeling (HLM), in which individual trials

(Stroop) were supposedly nested under participants, the *t*-test result reported for the interaction term between condition and willpower mindset—"*t*(1433) = 6.71, *p* < .01" (p. 1687)—suggests that their HLM analysis might not have properly coded participants as a random variable, instead treating all trials in the model (24 incongruent trials × 60 participants = 1,440 total trials) as though they came from 1,440 unique observations (see also [21]). Furthermore, their HLM analysis of their Stroop data contained additional anomalies. Specifically, Job et al. [1] conducted a multilevel logistic regression analysis on the binary-coded *accuracy* of an individual trial (incongruent trials only) on several variables (e.g., reaction times [RTs] for that trial), including some theoretically unmotivated covariates like age and trial order. Curiously, the Stroop data from subsequent ego-depletion studies from this group [19, 20] were analyzed quite differently, using different data-analytic methods, different dependent measures, and different covariates, even including lab rooms [19] in one study (see [22] for a description).

For all these statistical reasons, Schimmack [21] expressed skepticism about the replicability of Job et al.'s [1] original finding: "It is doubtful that a replication study would replicate the interaction between depletion manipulations and the implicit theory manipulation reported in Job et al. (2010) in an appropriate between-subject analysis." Thus, it seemed important to establish that Job et al.'s [1] results are replicable and are not tied to such unconventional ways of analyzing the data.

## The current study

We conducted a preregistered replication of Job et al.'s [1] Study 1, with a substantially bigger sample size (final *N* = 187), using the same two tasks (i.e., letter cancelation as the depletion task and Stroop as the outcome task). It is important to note, however, that, although we used the same tasks, we intentionally made several changes to strengthen the study design (e.g., administering manipulation checks, using a vocal-response version of Stroop, adding a second extension block for Stroop, removing what we perceived as an experimental confound), as will be elaborated and justified below.

Like Job et al. [1], we tested two hypotheses: (a) that the overall ego-depletion effect would be significant and, more importantly, (b) that this effect would be moderated by willpower mindset. We also evaluated whether a likely correlate of willpower mindset—trait self-control —would demonstrate a similar moderating influence. That is, if we successfully replicated the original study, our goal was to test the possibility that it was trait self-control, rather than willpower mindset per se, that might be the critical variable behind the moderating influence of willpower mindset on the ego-depletion effect.

## Method

All relevant experimental materials, data files, and data-analysis scripts (R codes) are available at the project OSF site (https://osf.io/37tgm/).

## Preregistration

Preregistration of this study was completed on October 18, 2016. The preregistration document is available on the Open Science Framework (OSF) at https://osf.io/e95bn. We note any deviations from the preregistered protocol where applicable below. Most of the procedural deviations occurred because we prematurely preregistered our study before the design and procedure were fully finalized, and we failed to update the preregistered protocol accordingly. All of the deviations from the preregistered protocol were minor.

## Participants

This research was approved by the University of Colorado Boulder Institutional Review Board (protocol #16–0033). Participants were recruited from the human subject pool of the Department of Psychology and Neuroscience, with data collection beginning in October 2016 and ending in May 2017. They received either partial course credit or payment for their participation. Individual-level data were anonymized by assigning arbitrary ID numbers to each participant. The authors therefore did not have access to any personally identifying information outside of the experimental sessions (Carruth facilitated some of the in-person experimental sessions).

After providing written informed consent, a total of 194 participants ($n = 97$ in each condition) took part in the study. They were randomly assigned to either the control or depletion condition. Prior to the study, we created a randomized list of condition assignments for the first 50 participants and repeated this step for each subsequent 50 participants.

Of the 194 participants, seven met one of the preregistered exclusion criteria and hence were excluded from the analyses:

- not finishing the study for various reasons, such as an experimenter ending a session early when a participant failed to follow instructions ($n = 1$ in the control condition); computer crashes occurring in the middle of the main Stroop task ($n = 1$ in the control condition and $n = 3$ in the depletion condition); and participants not completing key portions of questionnaires ($n = 2$ in the depletion condition);

- correctly guessing at least one of the hypotheses in the study ($n = 0$); and

- being a nonnative English speaker ($n = 0$).

Thus, the final sample consisted of 187 participants: $n = 95$ in the control condition (60 women) and $n = 92$ in the depletion condition (42 women).

Our preregistration document included two additional exclusion criteria: (a) missing more than one attention-check item in the questionnaire survey; and (b) being outside the range of 18 and 30 years old (an exclusion criterion used in Hagger et al.'s [11] multilab study, in which Carruth and Miyake participated). The first exclusion criterion was not applied because we ended up not including any attention check items in the questionnaires. We also decided against applying the age-based exclusion after learning that four participants chose not to provide their age (all in the depletion condition) and that three participants outside this age range were close to the upper-cutoff age (two 31-year-olds and one 32-year-old, all in the control condition). Statistical conclusions remained the same, however, even if these seven participants were excluded ($N = 180$; $n = 92$ in the control condition and $n = 88$ in the depletion condition).

## Sample size and power considerations

The original study [1] used ~30 participants per condition (total $N = 60$). Because of the way the data were analyzed and reported in the original study, however, it was not possible to derive effect-size estimates for the main effect of depletion or the moderation effect. Also, at the time of the preregistration (October 18, 2016), there were no similar published studies that administered the same willpower-mindset measure and reported its moderating influence on the ego-depletion effect. Thus, as specified in the preregistration, we followed what we had done before in our previous preregistered ego-depletion study [23] and tried to collect data from 100 participants in each condition.

Because we fell short of this sample-size goal (final $N = 187$) due to the end-of-semester closing of our subject pool, we conducted post-hoc power analyses using G*Power for the

primary preregistered analyses (referred to as the fixed-effects modeling below). With the current sample size, the effect size for the main effect of ego depletion that we could detect with 80% power was $d = .412$. Similarly, the effect size for the hypothesized moderation effect (i.e., an interaction between willpower mindset and depletion manipulation) with 80% power was $f^2 = .042$, which corresponds to $d = .410$. Thus, our study (at least our fixed-effects modeling reported below) was underpowered to detect smaller effects one may now expect on the basis of more recent large-sample multilab studies of ego depletion [11–13], even though our sample size was more than 3 times larger than that of the original Job et al. [1] study.

## Materials and procedure

Carruth (the lead author of this article) and undergraduate research assistants ran experimental sessions, and they all closely followed an experimenter procedural script. Except for Carruth, the experimenters were all blind to the hypotheses of the study.

Participants were tested individually in a quiet room, and the testing session lasted ~45–50 min. As we have done before [23], the experimenter was present in the room during the entire session but sat discreetly on the opposing side of a filing cabinet (blocking any view of the participant) while the participant completed the questionnaires and the letter cancelation task. During the Stroop task, the experimenter sat behind the participant so that they could record error trials (including voice-key errors) on a scoring sheet.

The overall procedure for this study is schematically illustrated in Fig 1. Although our procedure generally followed Job et al.'s [1], there were some notable changes that we introduced to strengthen the study design. We provide this figure to make clear the time course of administering manipulation-check measures and different types of trials across blocks.

The questionnaires and manipulation-check ratings were administered via Qualtrics. The Stroop task (both baseline and main blocks) were programmed with the PsyScope software (Version 1.2.5) and were run on an Apple computer (Mac OS X 10.6.4), using a button box capable of serving as a voice key.

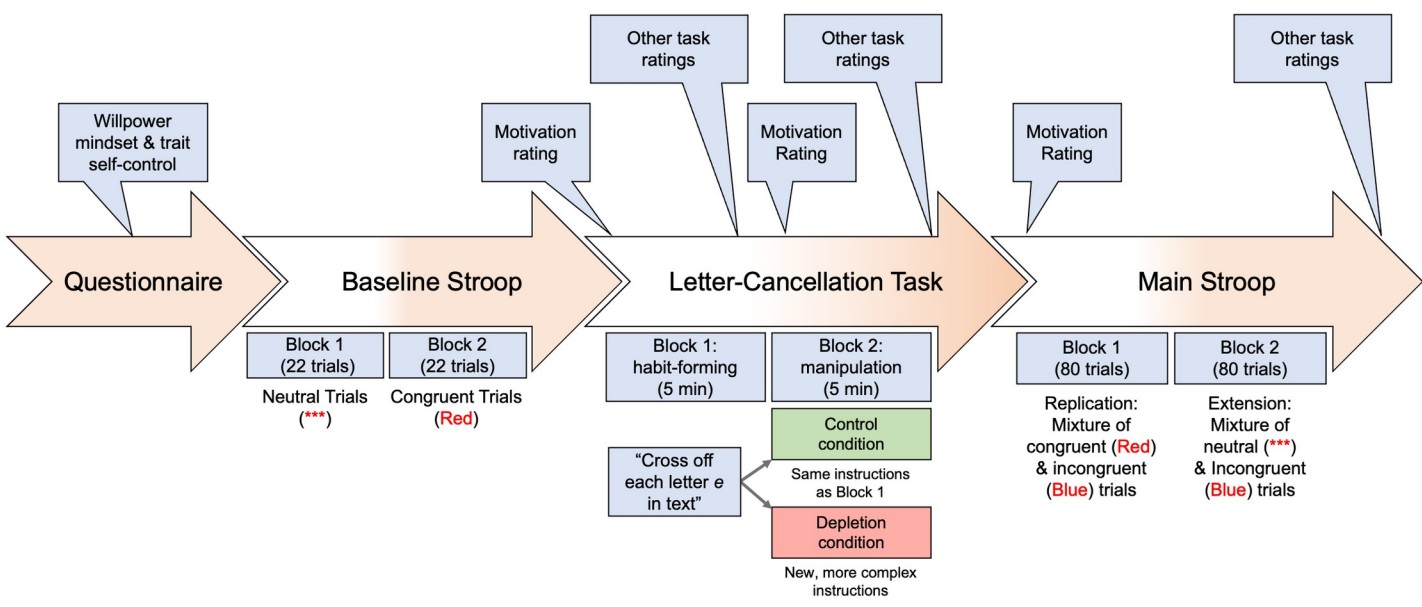

**Fig 1. A schematic summary of the experimental procedure.**

## Questionnaires

Participants first completed two questionnaires: (a) the Beliefs about the Willpower Scale [1] and (b) the full 36-item version of the Self-Control Scale [24]. Although our preregistration document specified that we planned to administer a susceptibility to ego-depletion questionnaire, this questionnaire was not administered due to time constraints. Additionally, the preregistration specified the inclusion of attention-check items in our questionnaire, but such items were not included due to the relatively short nature of the two questionnaires.

The Beliefs about the Willpower Scale consisted of two subscales: (a) beliefs about strenuous mental activities and (b) beliefs about resisting temptation. Although the original study (Study 1) [1] administered only the first subscale, we administered both subscales because one study reported in the original article used the full scale (Study 4). The Self-Control Scale [24], a popular trait self-control measure, was included to examine whether Job et al.'s moderation finding may have been driven by a likely correlate of willpower mindset such as trait self-control. Although Job et al. [1] indicated (in Footnote 4 on p. 1693) that their willpower-mindset measure did not correlate with trait self-control ($r = .17$, $p > .20$) in their pilot study ($N = 65$), it seemed important to evaluate this possibility in a more appropriately sized sample for correlational analysis. In fact, a subsequent study from this group [25] reported a more substantial correlation between these two measures ($r = -.47$, $N = 738$).

The two questionnaires were combined into a single survey, and the order of item presentation was randomized for each participant. For both questionnaires, participants rated each item on a 7-point Likert scale (1 = strongly disagree; 7 = strongly agree). We computed the average rating for each questionnaire and used it as a measure of willpower mindset and of trait self-control. A higher rating meant a stronger belief that willpower is not limited or a higher level of trait self-control, respectively. To avoid a potential problem of confusing participants by using inconsistent rating scales for the two questionnaires, we reversed the rating scale for the willpower-mindset questionnaire so that 1 meant strongly disagree and 7 strongly agree for both questionnaires. We also changed the rating scale for willpower mindset from a 6-point scale to a 7-point scale to make it comparable to that of the trait self-control measure (1 meant strongly agree and 6 strongly disagree in the original willpower-mindset scale). The reliability estimates of these measures (Cronbach's $\alpha$) were all satisfactory: .85 (the first subscale) and .77 (the entire scale) for willpower mindset and .89 for trait self-control.

## Baseline Stroop

Before the depletion manipulation, participants completed two brief Stroop trial blocks as baseline measures (not administered in Job et al.'s [1] study). We administered baseline blocks early to familiarize participants with the primary requirement of the Stroop task (saying aloud the *color* of each stimulus) and thereby minimize the lag between the two primary tasks (i.e., no need to provide detailed instructions after the administration of the depletion manipulation). Additionally, the experimenter used the baseline blocks to calibrate the sensitivity of the voice key and make necessary adjustments to the microphone. As we have consistently done in our work on executive functions [26–28], we used a vocal-response (rather than manual-response) version of the Stroop task, because vocal responses elicit a stronger need for overriding dominant responses than do manual responses [29] (i.e., saying "red" upon seeing the word "green" printed in red is more challenging than simply pressing a key that is arbitrarily mapped onto a particular color name).

In the first baseline block (Block 1), strings of 3–5 asterisks (\*\*\*, \*\*\*\*, or \*\*\*\*\*) were presented in red, blue, or green on a computer screen in a randomized order. Participants were asked to say aloud the color in which each string of asterisks was presented. There was an

initial set of 5 practice/calibration trials before moving on to 22 real baseline trials. In the second baseline block (Block 2), participants received 22 real trials of all congruent stimuli, where the word (*RED*, *BLUE*, or *GREEN*) matched the color it was presented in (e.g., *BLUE* presented in blue). Again, participants were asked to name the color that each word was presented in. Because these short baseline blocks did not include any incongruent trials, they should not have depleted the hypothesized self-control resources.

In all of the Stroop blocks (both baseline and main Stroop blocks), each trial began with a fixation cross (+), presented for 250 ms before the target stimulus appeared. The participant's vocal response to the stimulus terminated its presentation. A fixation cross signaling the beginning of the next trial appeared 750 ms after the participant's vocal response. This cycle continued until a brief rest period inserted in between blocks was reached.

The RTs were recorded by the program. Using a scoring sheet listing all the correct responses for the entirety of the Stroop trials, the experimenter scored the accuracy of each response as well as noting any voice-key error trials (e.g., trials on which the participant coughed or made some other noise, thus accidentally triggering the start of a subsequent trial).

### Depletion manipulation: Letter cancelation task

The task materials and the script for the letter cancelation task were obtained from Veronika Job. We used the same text and instructions, with one change noted below.

The letter cancelation task consisted of two blocks. In Block 1, participants in both conditions were asked to cross off letter *e*'s on a page of text with a pen. In this "habit-forming" block, participants received the following written instructions:

> "In this experiment, we are examining factors involved in stimulus identification and detection. The following page contains printed text. In the printed text, we want you to cross out selected text using the following rule: *cross out each letter "e" that appears in the text.*"

After reading these instructions, the participant responded to a motivation question, "How motivated are you to do well on this task?" (1 = not at all to 9 = very). Following this question, the experimenter instructed participants to begin working on the task, started a timer, and then instructed them to stop working after 5 min. The experimenter then administered the following manipulation-check questions, to which participants responded on the same 9-point scale (the underline/italics was used to highlight the dimension inquired by each question):

- How *effortful* did you find the task?

- How *tired* do you feel after doing the task?

- How *difficult* did you find the task?

- How *boring* did you find the task?

- How *frustrated* were you while you were doing the task?

- How much *effort* did you put into the task?

In Block 2, participants in the control condition were given instructions identical to Block 1 and were provided with a fresh page of text to work through (as with Job et al. [1], the text used for Block 2 was the same as that used for Block 1). In the depletion condition, however, participants received the following more complex instructions:

"In the printed text on the following page, we want you to cross out selected text using the following rule: cross out each letter "e" that appears in the text EXCEPT when another vowel (A, E, I, O, or U) <u>follows</u> the "e" in the same word (example: "READ") or when a vowel is one letter removed from the letter "e" in either direction in the same word (example: "VOWEL")."

After the experimenter made sure that participants understood the task requirements for Block 2, participants in both conditions were asked the same motivation question and then worked on Block 2 of the letter cancelation task for 5 min. After Block 2's completion, the experimenter provided participants with the same manipulation-check questions administered after Block 1.

Important to note, in the Job et al. [1] study, the page of text given to the depletion condition in Block 2 differed in visual appearance from that used in the control condition. Specifically, the text itself was blurred in the depletion condition, but not in the control condition. Although the rationale for this difference was not mentioned in the original article, Job et al. [1] explicitly stated that the goal of the letter cancelation task was to "establish a behavioral pattern" in Block 1 and then to require participants in the depletion condition to "inhibit the previously established response" in Block 2 (p. 1687; see also [30] for the same argument). Given this stated purpose of the letter cancelation task, we decided that using a blurred text in the depletion condition was not an integral component of this task (i.e., perceptual degradation of the text would unlikely increase the hypothesized self-control demand of "inhibiting the previously established response"). For this reason, we opted for not using a blurred version of text in the depletion condition and thereby avoiding an unnecessary experimental confound.

### Outcome measure: Main Stroop task

Participants next completed the main Stroop task, which, like the earlier baseline task, also consisted of two blocks. First, as in Job et al. [1], participants completed a block of mixed congruent and incongruent trials. This replication block (Block 1) was longer than that in the original Job et al. [1] study (24 congruent and 24 incongruent trials) and consisted of two subblocks of 40 trials each for a total of 80 trials (40 congruent and 40 incongruent trials), with a brief break in between (our preregistration incorrectly specified two blocks of 72 trials each instead of 80 trials each). We increased the number of Stroop trials from 48 [1] to 80, because the Stroop interference effect is highly susceptible to occasional long RTs. After receiving a brief reminder emphasizing the importance of naming the color (not reading the word) and performing 6 practice trials, participants responded to the same motivation question before receiving the experimental trials.

As part of the planned deviations from Job et al.'s [1] procedure, we added a second block of mixed incongruent and neutral (asterisk) trials (Block 2). Again, this extension block consisted of two subblocks of 40 trials each (for a total of 80 trials, split evenly between neutral and incongruent). Other than using a different mixture of trial types, the trials were set up in the same way as in Block 1.

We added this extension block because prior research has suggested that the 50/50 mixture of congruent and incongruent trials (as was done in the original study [1]) has been known to elicit a phenomenon known as goal neglect [31] (see also [32]), a temporary loss of the task goal (i.e., color naming) from working memory due to the abundance of cues supporting an inappropriate task goal (in this case, word reading, due to the large number of congruent trials for which word reading suffices). Thus, the way the Stroop task was administered in the original study critically hinged on working memory maintenance of task goals. In contrast, a 50/50

mixture of incongruent and neutral (asterisk) trials used in Block 2 would make the Stroop task more appropriate as a measure of overriding dominant yet inappropriate responses.

After the completion of Block 2 for the Stroop task, participants responded to the same self-rating questions administered at the end of both blocks of the letter cancelation task (effortful, tired, etc.), although those post-Stroop ratings were not of primary interest. Finally, participants responded to a single open-ended question asking what they thought might be the purpose of the study and provided demographic information (age and gender) before they were debriefed and dismissed.

### Data-exclusion and data-transformation considerations

For the analyses of the baseline and main Stroop data, the calculation of error rates (%) excluded voice-key error trials noted by the experimenter (e.g., voice key was triggered due to a cough or a participant saying "uh. . ."). The analyses of the RT data excluded voice-key error trials as well as actual error trials. Additionally, we excluded an extremely small number of RT trials (.03% of all trials) associated with impossible RTs (< 0 ms), which likely happened when a participant's vocalization for a particular trial started just about the same time as the appearance of the Stroop stimulus. We also performed the fixed-effects and mixed-effects modeling reported below after excluding extreme RTs that are unlikely to reflect normal cognitive processes associated with Strop performance (< 200 ms and > 5,000 ms). Removing these rare trials (.09% of all trials) did not change any statistical conclusions.

Given that the Stroop data are noisy, often contain outliers (especially in the RT data), and show skewed distributions (both error and RT data), our preregistered plan was to analyze the data with and without transformations and make sure that the results are robust to different transformations. Thus, besides using raw error rates and RTs, we also analyzed the data after applying (a) an arcsine transformation for accuracy (proportion accuracy scores, namely "1 minus error rate") and (b) a logarithmic (natural log) transformation for RTs. For simplicity, the reporting of our results below focuses on the raw error rates and the log-transformed RTs. The conclusions essentially remain the same, however, when the analyses used arcsine accuracy or raw RT data (see Tables A and B in S1 Appendix). We had also planned to conduct the RT analyses using Wilcox-Keselman trimming procedure, but, because the raw RT and log RT results converged, we decided to skip this extra RT analysis.

## Results

No analyses of the data were conducted until after the data collection was completed. An alpha level for statistical tests was set at .05. All the analyses reported in this article were conducted with the base R statistical packages, except for mixed-effects models, which were computed using the lme4 package in R. All the relevant data and the annotated R codes used for the analyses are available at our project OSF site (https://osf.io/37tgm/).

### Preliminary analyses

**Descriptive statistics.**   The descriptive statistics for age, the questionnaire measures, and baseline Stroop performance are summarized in Table 1, separately for each condition. Also shown in the table are the results of independent-sample *t* tests comparing the two conditions.

Random assignment was generally successful, and none of the measures showed significant differences, except for the log RT for Block 2 of the baseline Stroop task for the congruent trials. This small but significant RT difference was unexpected, but the log RT data for the neutral baseline trials did not show such a group difference, thus ruling out the possibility of preexisting general processing-speed differences between the two groups.

**Table 1. Descriptive statistics for demographic and baseline measures.**

| | Control | | | Depletion | | | | |
|---|---|---|---|---|---|---|---|---|
| | *n* | *M* | *SD* | *n* | *M* | *SD* | *t* | *p* |
| **Demographics** | | | | | | | | |
| Age | 95 | 19.64 | 2.81 | 88[a] | 19.31 | 1.48 | 0.998 | 0.319 |
| **Individual difference scales** | | | | | | | | |
| Willpower mindset: SMA | 95 | 3.22 | 1.00 | 92 | 3.27 | 1.01 | −0.382 | 0.703 |
| Willpower mindset: Full Scale | 95 | 3.89 | 0.76 | 92 | 3.98 | 0.63 | −0.875 | 0.383 |
| Trait self-control | 95 | 4.14 | 0.69 | 92 | 4.18 | 0.69 | −0.394 | 0.694 |
| **Baseline Stroop measures** | | | | | | | | |
| Error rate for asterisk trials | 95 | 0.39 | 1.60 | 91[b] | 0.16 | 1.10 | 1.164 | 0.246 |
| Log RT for asterisk trials | 95 | 6.36 | 0.13 | 91[b] | 6.38 | 0.12 | −1.403 | 0.162 |
| Error rate for congruent trials | 95 | 0.05 | 0.47 | 91[b] | 0.05 | 0.48 | −0.030 | 0.976 |
| Log RT for congruent trials | 95 | 6.27 | 0.14 | 91[b] | 6.31 | 0.13 | −2.181 | **0.030** |

*Note*. SMA = the strenuous mental activity subscale (1 = limited willpower, 7 = nonlimited willpower). Willpower mindset: Full scale was computed by taking an average of the first SMA subscale and the second "resisting temptations" subscale. The *t* values and *p* values are results from independent-sample *t* tests comparing differences between participants in the control condition and participants in the depletion condition.

*a*. Four participants (all in the depletion condition) did not indicate their ages.

*b*. The baseline Stroop measure from one participant in the depletion condition was missing because of a computer crash that occurred during this portion of the study (we kept this participant's data in because the rest of the study went fine).

## Manipulation checks

We conducted manipulation checks—a step missing in the original Job et al. [1] study—by testing whether the control and depletion conditions differed in the self-ratings provided before (motivation) and after each block of the letter cancelation task (effortful, tired, difficult, boring, frustrated, and effort). As noted earlier (see Fig 1), the self-ratings were solicited for both the initial letter cancelation block common to all participants (Block 1) and the manipulation block that differed in the two conditions (Block 2).

We expected no group differences in the motivation ratings collected before both blocks of the letter cancelation task, because, at the beginning of Block 2, the control and depletion groups had not diverged yet. More importantly, although we did not expect any condition differences in the other ratings collected at the end of Block 1, we expected systematic differences in the ratings collected after Block 2. These expectations were generally confirmed, except for tired, as described below.

*Non-motivation ratings*. The means (*M*s) and standard deviations (*SD*s) of the non-motivation ratings for each block (Block 1 and Block 2) are summarized in Table 2 below, along with the results of independent-sample *t*-test results.

As shown in Table 2, one measure (tired) produced unexpected results for both blocks (significant after Block 1 but nonsignificant after Block 2). Thus, we conducted an exploratory follow-up analysis. Specifically, for each participant and for each rating, we computed a difference between the two blocks (i.e., Block 2 rating minus Block 1 rating) so that the resulting number would indicate a change from Block 1 to Block 2. Higher difference scores meant a perceived increase in individual ratings from Block 1 to Block 2. We then compared the resulting difference scores for each rating between the control and depletion conditions. Note that this analysis is essentially the same as conducting a 2 × 2 mixed ANOVA (with condition as a between-subjects variable and blocks as a within-subjects variable) for each rating and

**Table 2. Participant ratings after Blocks 1 and 2 of Task 1, and within-person differences between tasks.**

| | Control | | | Depletion | | | | |
|---|---|---|---|---|---|---|---|---|
| | *n* | *M* | *SD* | *n* | *M* | *SD* | *t* | *p* |
| **After Block 1** | | | | | | | | |
| Effortful | 95 | 5.77 | 2.16 | 92 | 5.93 | 2.03 | 0.543 | 0.588 |
| Tired | 95 | 4.29 | 2.19 | 92 | 3.50 | 1.92 | −2.635 | **0.009** |
| Difficulty | 95 | 3.20 | 1.90 | 92 | 3.27 | 1.72 | 0.270 | 0.787 |
| Boring | 95 | 5.72 | 2.21 | 92 | 5.29 | 2.31 | −1.279 | 0.202 |
| Frustration | 95 | 2.60 | 1.94 | 92 | 2.40 | 1.77 | −0.728 | 0.468 |
| Effort | 95 | 6.84 | 1.76 | 92 | 7.02 | 1.70 | 0.709 | 0.479 |
| **After Block 2** | | | | | | | | |
| Effortful | 95 | 5.30 | 2.35 | 92 | 6.95 | 1.52 | 5.650 | **0.000** |
| Tired | 95 | 4.67 | 2.26 | 92 | 4.46 | 2.07 | −0.685 | 0.494 |
| Difficulty | 95 | 3.41 | 1.97 | 92 | 5.91 | 1.83 | 8.988 | **0.000** |
| Boring | 95 | 5.67 | 2.52 | 92 | 5.47 | 2.37 | −0.575 | 0.566 |
| Frustration | 95 | 3.00 | 1.97 | 92 | 4.00 | 2.21 | 3.271 | **0.001** |
| Effort | 95 | 6.60 | 1.94 | 92 | 7.24 | 1.49 | 2.521 | **0.013** |
| **Rating Difference (Block 2 –Block 1)** | | | | | | | | |
| Effortful | 95 | −0.46 | 1.31 | 92 | 1.01 | 1.57 | 6.971 | **0.000** |
| Tired | 95 | 0.38 | 1.25 | 92 | 0.96 | 1.30 | 3.100 | **0.002** |
| Difficulty | 95 | 0.21 | 1.30 | 92 | 2.64 | 1.63 | 11.316 | **0.000** |
| Boring | 95 | −0.04 | 2.06 | 92 | 0.17 | 1.85 | 0.754 | 0.452 |
| Frustration | 95 | 0.40 | 1.45 | 92 | 1.60 | 1.55 | 5.455 | **0.000** |
| Effort | 95 | −0.24 | 1.11 | 92 | 0.22 | 1.05 | 2.913 | **0.004** |

*Note.* The *t* values and *p* values reported are results from independent-sample *t*-tests examining comparing group differences between participants in the control condition and participants in the depletion condition.

testing for the two-way interaction effect. The results of this comparison are summarized in the bottom panel of Table 2 and are visually illustrated in Fig 2.

For all the non-motivation ratings collected after Block 2, we expected a significant difference between the two conditions, and this is exactly what we observed, with the exception of boring, $t(185) = −.754$, $p = .452$. Important to note, although the tired ratings produced unexpected results when the Block 1 and Block 2 ratings were analyzed separately, this difference-score analysis produced the result consistent with our prediction: An increase in the self-perceived level of tiredness from Block 1 to Block 2 was significantly greater among participants in the depletion condition (from 3.50 to 4.46) than among those in the control condition (from 4.29 to 4.67).

To further confirm the overall effectiveness of the main depletion manipulation, we conducted a one-way MANOVA (not initially planned) that simultaneously took into account all six ratings and used the difference scores for the respective ratings as the dependent measures. The MANOVA results indicated a significant difference between the control and depletion conditions, Pillai-Bartlett trace $V = 0.470$ ($df = 1$), approximate $F(6, 180) = 26.55$, $p < .001$, suggesting that our manipulation checks were successful, even though one of the six ratings, boring (arguably the least directly relevant to the hypothesized "depleted" state), did not yield an expected pattern. (Parenthetically, we note here that the difference-score measures used in our exploratory MANOVA were mostly significantly correlated with each other, generally in the .20 to .50 range, except for boring, which did not correlate with the other ratings).

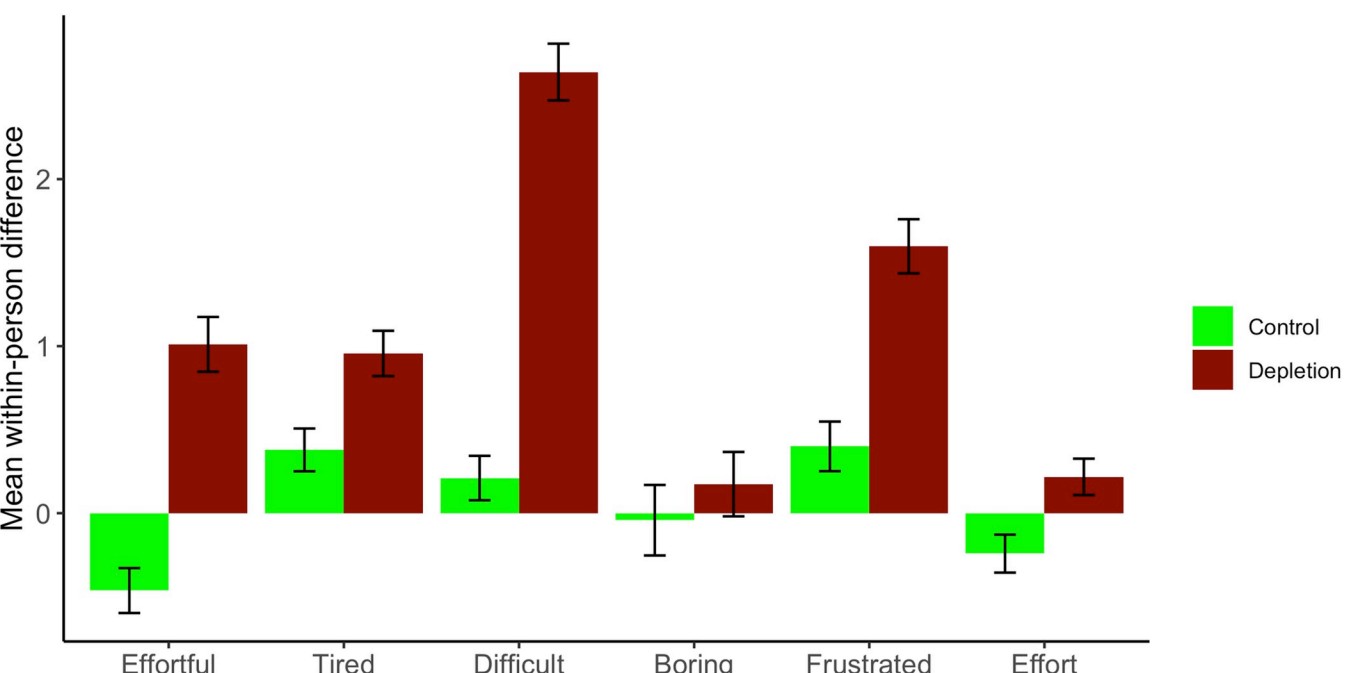

**Fig 2. Mean within-person differences in ratings between Block 1 (the habituation block) and Block 2 (the critical manipulation block) for the letter cancelation task.** Higher bars represent higher ratings after Task 2, relative to Task 1. Error bars represent standard errors.

*Motivation ratings.* As illustrated in Fig 1, the two experimental conditions (control vs. depletion) had not diverged when the initial motivation ratings were obtained (i.e., at the outset of the habituation block, Block 1, as well as the manipulation block, Block 2). Accordingly, there should not be any condition differences for those motivation ratings. Consistent with this expectation, the self-reported levels of motivation for performing Block 1 of the letter cancelation task did not significantly differ in the two conditions ($M = 7.57$ and $SD = 1.25$ in the control condition and $M = 7.51$ and $SD = 1.40$ in the depletion condition), $t(185) = -0.297$, $p = .767$.

The same was true for the levels of motivation before performing Block 2 of the same task ($M = 7.12$ and $SD = 1.56$ in the control condition and $M = 7.04$ and $SD = 1.72$ in the depletion condition), $t(185) = -0.301$, $p = .763$. Moreover, the degree of rating changes between Block 1 and Block 2 (operationalized as Block 2 ratings minus Block 1 ratings) did not show any condition differences, $t(185) = -0.840$, $p = .933$.

### Primary preregistered analyses: Fixed-effects models

We first conducted our preregistered analyses (referred to as the "fixed-effects analysis" here), which treated only participants (but not trials/stimuli) as a random variable. Specifically, for both error and RT data, we first computed mean error rates and RTs for each within-subject condition (congruent vs. incongruent) for each participant. Then, using the Stroop interference score (operationalized as "incongruent minus congruent trials") as the dependent measure, we conducted independent-sample $t$ tests to test whether there was any evidence for the ego-depletion effect at the overall group level in each block. This analysis was followed up by regression analyses to test the hypothesized interaction effect between condition (depletion vs. control) and willpower mindset (treated as a continuous variable).

**Table 3. Descriptive statistics for Stroop performance measures in Blocks 1 and 2.**

|  | Control | | | Depletion | | | | |
|---|---|---|---|---|---|---|---|---|
|  | N | *M* | SD | N | *M* | SD | t | p |
| **Stroop Task Block 1** | | | | | | | | |
| Error rate for congruent trials | 95 | 0.30 | 0.98 | 92 | 0.56 | 1.69 | 1.300 | 0.195 |
| Log RT for congruent trials | 95 | 6.54 | 0.16 | 92 | 6.56 | 0.13 | 0.978 | 0.329 |
| Error rate for incongruent trials | 95 | 4.77 | 7.37 | 92 | 3.67 | 3.97 | −1.262 | 0.209 |
| Log RT for incongruent trials | 95 | 6.71 | 0.16 | 92 | 6.72 | 0.15 | 0.752 | 0.453 |
| Difference in error rates (Inc–Cong) | 95 | 4.47 | 6.97 | 92 | 3.12 | 4.09 | −1.620 | 0.107 |
| Difference in Log RT (Inc–Cong) | 95 | 0.17 | 0.08 | 92 | 0.16 | 0.07 | −0.360 | 0.719 |
| **Stroop Task Block 2** | | | | | | | | |
| Error rate for asterisk trials | 95 | 0.69 | 1.44 | 92 | 0.72 | 1.53 | 0.136 | 0.892 |
| Log RT for asterisk trials | 95 | 6.50 | 0.13 | 92 | 6.53 | 0.13 | 1.352 | 0.178 |
| Error rate for incongruent trials | 95 | 3.47 | 5.32 | 92 | 2.99 | 4.15 | −0.681 | 0.497 |
| Log RT for incongruent trials | 95 | 6.67 | 0.15 | 92 | 6.70 | 0.15 | 0.994 | 0.322 |
| Difference in error rates (Inc–Neut) | 95 | 2.78 | 5.20 | 92 | 2.27 | 4.33 | −0.721 | 0.472 |
| Difference in Log RT (Inc–Neut) | 95 | 0.17 | 0.06 | 92 | 0.17 | 0.07 | −0.366 | 0.715 |

*Note*. Inc = Incongruent trials; Cong = Congruent trials; Neut = Neutral (asterisk) trials. The *t* values and *p* values are results from independent-sample *t* tests comparing differences between participants in the control condition and participants in the depletion condition.

## Testing the main effect of ego depletion

The condition means as well as Stroop interference scores for the error data (error proportion) and RT data (log RTs) are presented in Table 3. To visually illustrate the data distribution, we also provided the raincloud plots for the error and log RT data for both Stroop blocks in Figs A–C in S1 Appendix. As is clear from these plots, the Stroop interference effect was present, especially for the RT data, but, for neither the replication block nor the extension block was there any evidence that the Stroop effect was greater in the depletion condition than in the control condition.

*Block 1.* As summarized in the top panel of Table 3, a *t* test for the Stroop interference effect for the error data failed to reveal a significant difference between the two conditions, $t(185) = -1.620$, $p = .107$, $d = -0.239$, 95% CI for the condition difference [−3.014, .296]. Similarly, the condition difference for the Stroop interference effect was not significant for the log RT data, either, $t(185) = -0.360$, $p = .719$, $d = -0.053$, 95% CI for the condition difference [−.025, .017].

As noted earlier, our Block 1 Stroop included substantially more trials (a total of 80, administered in two subblocks of 40 trials each) than Job et al.'s [1] study did (a total of 48). Because one could argue that the impact of the depletion manipulation might have faded quickly over the course of the Stroop task, we reanalyzed our data, focusing only on the first 40 trials of the replication block (Block 1) so that the number of Stroop trials in the current study would be comparable to that of the Job et al. study. The results remained the same, however, as reported in Tables C–E in S1 Appendix.

Note that, as illustrated in Fig A in S1 Appendix, there was one extreme outlier (61.5% error rate) in the control condition (the next highest error rate observed in either condition was about 20%), which might have potentially influenced the *t*-test results for the Block 1 error data. However, even when this outlier was excluded, the condition difference in the Stroop interference effect in error rates (*M*s = 3.92 vs. 3.12) was still nonsignificant, $t(184) = -1.286$, $p = .200$, $d = -.189$, 95% CI for the condition difference [−2.047, .431].

*Block 2.* The extension block data showed the same pattern (see the bottom panel of Table 3). The Stroop interference effect did not differ between the two conditions, for the error

data, $t(185) = -.721$, $p = .472$, $d = -0.106$, 95% CI for the condition difference [−1.890, .878], as well as for the log RT data, $t(185) = -.366$, $p = .715$, $d = -0.053$, 95% CI for the condition difference [−.023, .016].

## Testing the moderation effect

Next, we tested whether willpower mindset would moderate the ego-depletion effect by conducting regression analyses in which the dependent variable (the Stroop interference effect) was regressed onto three variables: condition (control vs. depletion; coded as −1 and +1, respectively), willpower mindset (treated as a continuous variable), and the interaction between the two. The results are summarized in Table 4.

*Block 1.* There was no evidence for the hypothesized moderation effect for the RT data in this replication block: The *B* estimate for the interaction term was .001, $p = .778$. For the error data, however, the interaction term was significantly different from zero, $B = .875$, $p = .038$, suggesting the presence of the moderating influence of the ego-depletion effect by willpower mindset, as was demonstrated by Job et al. [1].

A close look at the data revealed, however, that this interaction effect did not qualify as a replication of Job et al.'s [1] finding because the directionality of the moderation effect was the opposite of what they would predict. Specifically, as illustrated in Fig 3, in our data, individuals with a *nonlimited*-willpower mindset were associated (albeit marginally) with *lower* error rates among participants in the control condition (simple effect: $B = -1.07$, $p = .072$), but not among those in the depletion condition (simple effect: $B = .683$, $p = .250$). Moreover, this opposite interaction effect was influenced by the extreme outlier in the control condition noted earlier (see Fig 3 right panel; the outlier is highlighted in that figure). In fact, when this extreme outlier was removed from analysis, the opposite interaction was no longer significant: $B = .444$ (95% CI [−.179, 1.066]), $p = .162$. As these additional results suggest, our primary analyses for Stroop Block 1 did not yield evidence for the proposal that only individuals with a limited-willpower mindset are susceptible to the ego-depletion effect.

*Block 2.* As shown in Table 4, there was no evidence for the hypothesized moderation effect for either dependent measure in the extension block. The unstandardized *B* estimate for the

**Table 4. Results from fixed-effect regression models for Stroop Blocks 1 and 2.**

|  | Stroop Interference (Error) | | | | Stroop Interference (Log RTs) | | | |
|---|---|---|---|---|---|---|---|---|
|  | *B* | *SE* | *95% CI* | *p* | *B* | *SE* | *95% CI* | *p* |
| **Block 1** | | | | | | | | |
| Intercept | 3.771 | 0.417 | [2.948, 4.593] | **<0.001** | 0.166 | 0.005 | [0.156, 0.177] | **<0.001** |
| Condition (C) | −0.674 | 0.417 | [−1.497, 0.148] | 0.107 | −0.002 | 0.005 | [−0.013, 0.009] | 0.736 |
| Mindset (M) | −0.191 | 0.418 | [−1.016, 0.633] | 0.648 | −0.004 | 0.005 | [−0.015, 0.007] | 0.455 |
| C × M | 0.875 | 0.418 | [0.051, 1.699] | **0.038** | 0.001 | 0.005 | [−0.009, 0.012] | 0.823 |
| **Block 2** | | | | | | | | |
| Intercept | 2.519 | 0.350 | [1.829, 3.209] | **<0.001** | 0.170 | 0.005 | [0.160, 0.179] | **<0.001** |
| Condition (C) | −0.238 | 0.350 | [−0.928, 0.452] | 0.498 | −0.002 | 0.005 | [−0.011, 0.008] | 0.721 |
| Mindset (M) | −0.547 | 0.351 | [−1.239, 0.145] | 0.120 | −0.001 | 0.005 | [−0.011, 0.009] | 0.840 |
| C × M | 0.287 | 0.351 | [−0.405, 0.979] | 0.414 | −0.003 | 0.005 | [−0.013, 0.007] | 0.558 |

*Note*. Stroop interference = the error rate or log RT difference between the incongruent trials and the congruent (Block 1) or neutral/asterisk (Block 2) trials. *B* = Regression parameters, not standardized with respect to the dependent variable. *SE* = Standard errors for *B*. *CI* = 95% confidence intervals for *B* estimates. Condition (between-subjects) was coded as −1 for the control condition and 1 for the depletion condition. Mindset (the first subscale of the willpower mindset measure) was mean-centered and standardized.

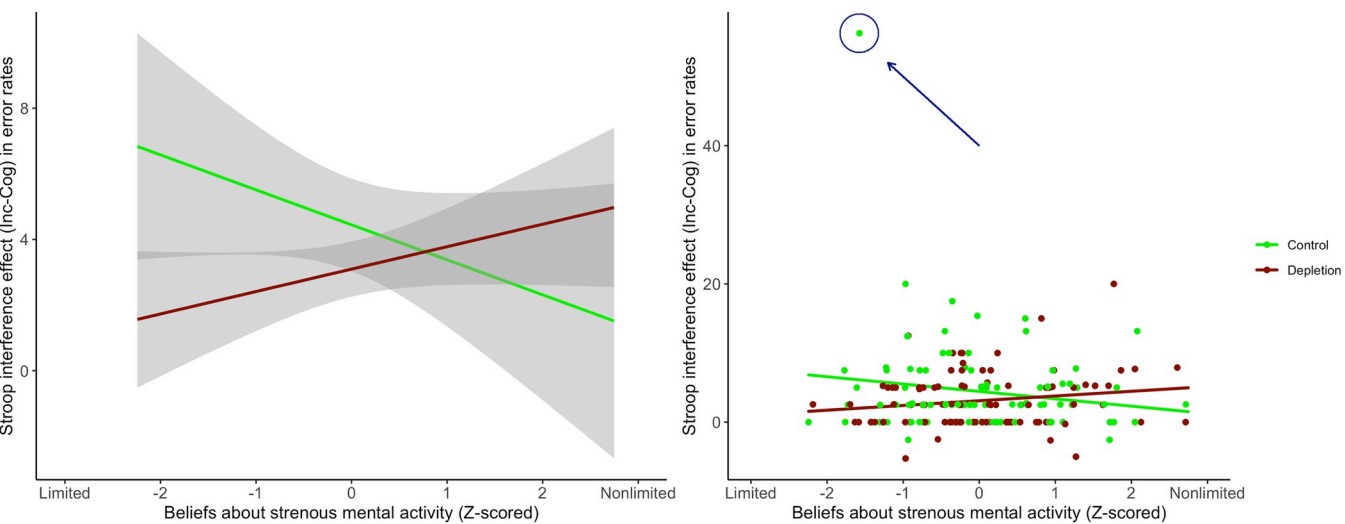

**Fig 3. A visual plot of the Condition × Mindset interaction for the Block 1 error data (left panel) with data points (right panel).** (Left Panel) The y axis represents the Stroop interference effect, computed as the percentage of errors made in incongruent trials minus the percentage of errors made in congruent trials. The curves around the lines represent 95% confidence intervals. Notably, the significant interaction pattern plotted here was the opposite of what was reported in the original Job et al. study; specifically, among participants in the ego-depletion condition, a nonlimited willpower mindset was associated with a higher Stroop interference effect (i.e., worse self-control performance). (Right Panel) As highlighted in the plot, there was a clear outlier on the top-left corner (also visible in the raincloud plot provided in Fig A in S1 Appendix). When this extreme outlier was removed from analysis, the interaction becomes nonsignificant.

interaction term was not significantly different from zero, for either the error data, $B = .287$, $p = .414$, or for the log RT data, $B = -.003$, $p = .558$.

## Preregistered follow-up analyses: Replicating Job et al.'s analyses

Upon failing to replicate Job et al.'s [1] main findings with our fixed-effects analyses, our preregistered follow-up plan was to analyze our data in the same way as Job et al. [1] did. Briefly stated, their analyses focused on incongruent trials only and conducted a multilevel (HLM) logistic-regression analysis at the level of individual trials (as noted earlier, however, trials did not seem to be properly nested under participants). The accuracy on a particular trial—binary coding (0 vs. 1) of "correct" vs "incorrect"—was regressed onto multiple trial-level variables (condition [control vs. depletion]; mean-centered trial numbers; and RT for that trial) and participant-level variables (willpower mindset and age, both mean-centered), along with an interaction term for condition and willpower mindset (see Job et al.'s [1] description on p. 1687).

We tried to replicate this unconventional analysis, but were not successful in our attempts, because the hierarchical logistic regression models that we implemented in R did not converge when RTs were simultaneously included in the model. Thus, we decided to analyze our data by testing more standard mixed-effects models instead, as described next.

## Exploratory analyses: Mixed-effects models

A major advantage of mixed-effects analyses is that they allow us to control for random variance between participants while also accounting for non-heteroscedasticity in the data. Additionally, mixed-effects models can allow the slopes of trial type (congruent vs. incongruent) to vary between participants and control for nuisance variance associated specifically with individual trials/stimuli, thus increasing the power to detect the hypothesized effects [33]. Thus, we conducted mixed-effects analyses, modeling both individual participants and trials simultaneously as random variables.

**Table 5. Results from mixed-effect regression models for Stroop Blocks 1 and 2.**

| Predictors | Odds of Incorrect Response | | | | Log RT | | | |
|---|---|---|---|---|---|---|---|---|
| | OR | SE | 95% CI | p | B | SE | 95% CI | p |
| **Block 1** | | | | | | | | |
| Intercept | 0.003 | 0.458 | [0.001, 0.008] | **<0.001** | 6.632 | 0.012 | [6.609, 6.655] | **<0.001** |
| Condition (C) | 1.094 | 0.176 | [0.775, 1.544] | 0.610 | 0.010 | 0.010 | [−0.010, 0.031] | 0.331 |
| Mindset (M) | 1.172 | 0.172 | [0.837, 1.642] | 0.356 | −0.028 | 0.010 | [−0.048, −0.007] | **0.009** |
| Trial Type (T) | 7.827 | 0.451 | [3.231, 18.958] | **<0.001** | 0.083 | 0.006 | [0.071, 0.095] | **<0.001** |
| C × M | 1.109 | 0.171 | [0.793, 1.550] | 0.545 | −0.006 | 0.010 | [−0.027, 0.014] | 0.544 |
| C × T | 0.818 | 0.160 | [0.597, 1.120] | 0.210 | −0.001 | 0.003 | [−0.006, 0.004] | 0.746 |
| M × T | 0.896 | 0.156 | [0.660, 1.216] | 0.481 | −0.002 | 0.003 | [−0.007, 0.003] | 0.492 |
| C × M × T | 1.080 | 0.155 | [0.797, 1.462] | 0.619 | 0.001 | 0.003 | [−0.005, 0.006] | 0.832 |
| **Block 2** | | | | | | | | |
| Intercept | 0.008 | 0.183 | [0.005, 0.011] | **<0.001** | 6.601 | 0.011 | [6.579, 6.622] | **<0.001** |
| Condition (C) | 0.978 | 0.109 | [0.790, 1.212] | 0.842 | 0.012 | 0.010 | [−0.007, 0.031] | 0.213 |
| Mindset (M) | 1.216 | 0.105 | [0.989, 1.494] | 0.063 | −0.017 | 0.010 | [−0.036, 0.002] | 0.083 |
| Trial Type (T) | 2.346 | 0.173 | [1.673, 3.291] | **<0.001** | 0.085 | 0.006 | [0.074, 0.096] | **<0.001** |
| C × M | 1.032 | 0.104 | [0.841, 1.265] | 0.765 | −0.011 | 0.010 | [−0.030, 0.009] | 0.283 |
| C × T | 0.968 | 0.096 | [0.801, 1.168] | 0.732 | −0.001 | 0.002 | [−0.006, 0.004] | 0.738 |
| M × T | 0.796 | 0.092 | [0.665, 0.953] | **0.013** | −0.001 | 0.002 | [−0.005, 0.004] | 0.814 |
| C × M × T | 1.035 | 0.091 | [0.866, 1.237] | 0.708 | −0.001 | 0.002 | [−0.006, 0.003] | 0.580 |

*Note*. *OR* = Unstandardized odds ratios. *B* = Regression parameters, not standardized with respect to the dependent variable. *SE* = Standard errors for *OR* and *B*. *95% CI* = 95% confidence intervals for *B* estimates. Condition (between-subjects) was coded as −1 for the control condition and 1 for the depletion condition. Mindset (the first subscale) was mean-centered and standardized. Trial type was coded as (−1 for congruent trials in Block 1 and neutral trials in Block 2 and 1 for incongruent trials in both Blocks 1 and 2). *P* values for binomial logistic mixed-effects models (for the error data) were calculated using Wald's *Z* statistic as estimated by the lme4 package in R. *P* values for linear mixed-effects models (for the log RT data) are calculated by lme4 package in R, based on Satterthwaite's approximations to degrees of freedom.

Specifically, we conducted four separate sets of mixed-effects regression models (for errors and log RTs, run separately for Stroop Blocks 1 and 2). For error data, we tested mixed-effects logistic regression models estimating the odds of making an error (coded as 0 for correct trials and 1 for error trials). For log RT data, we tested a mixed-effects linear regression model estimating log-transformed RTs. Both models included Condition (−1 = control vs. +1 = depletion) and willpower mindset (*z* scored) as between-subjects predictors and Trial Type (−1 = congruent trials in Block 1 and neutral trials in Block 2 vs. +1 = Incongruent) as a within-subjects predictor. For all models, we estimated, for each participant, random intercepts as well as random slopes for Trial Type (congruent vs. incongruent/neutral). Thus, the mixed-effects analyses we report here are comparable to those conducted by Savani and Job [20], although we did not include any additional covariates in our models. The results of these mixed-effects analyses are summarized in Table 5.

*Block 1*. As shown in Table 5 (top panel), the mixed-effects modeling results are consistent with fixed-effects modeling results reported earlier. For both error and RT data, the effect of Trial Type (i.e., the overall Stroop interference effect) was significant, as expected. More interesting is the finding that, for the log RT (but not error) data, willpower mindset was a significant predictor of the overall Stroop RTs in the expected direction (i.e., a nonlimited willpower mindset was associated with faster RTs, regardless of the types of trials). In neither analysis for Block 1, however, did willpower mindset interact significantly with Trial Type, Condition (depletion vs. control), or both.

*Block 2*. The mixed-effects modeling results for the Block 2 data were similar to those of Block 1 (bottom panel of Table 5). Again, as expected, there was clear evidence for the effect of Trial Type (neutral vs. incongruent) for both dependent measures. However, the hypothesized willpower-mindset moderation effect, namely the three-way interaction term (Condition × Mindset × Trial Type), was not significant for either error or log RT data. Interesting to note, however, a two-way interaction involving willpower mindset (Mindset × Trial Type) was significant for the error data, odds ratio (OR) = 0.796, *p* = .013, suggesting that the limited-willpower mindset was associated with the likelihood of producing larger Stroop interference effects (i.e., greater errors for incongruent trials than for neutral trials). Critically, however, this interaction effect did not involve the ego-depletion manipulation.

Because none of these mixed-effects models we tested provided any evidence of the moderating influence of willpower mindset on the ego-depletion effect hypothesized by Job et al. [1], we conducted one additional set of exploratory mixed-effects modeling. Specifically, in these further exploratory analyses, we specifically controlled for random effects of stimuli (e.g., specific combinations of words and colors) by additionally estimating random intercepts for the character set and color of the stimuli, as was done before by Meier and Kane [34]. The results remained the same for these additional models.

### Secondary (exploratory) analyses: Analyses based on the trait self-control measure

At the time of preregistration, we hypothesized that individual differences in willpower mindset would likely be correlated with other individual differences variables, such as trait self-control. Consistent with this expectation, the correlation between willpower mindset (the first subscale) and trait self-control in this study was small but significant, *r*(185) = .255, *p* < .001. If an individual's willpower mindset is systematically related to their perception of how good they are at self-control, it seemed possible that any moderation of ego depletion by willpower mindset might be driven primarily by trait self-control, rather than willpower mindset per se.

Because we did not find any evidence for the willpower-mindset-moderation hypothesis in the current study, however, we did not conduct our planned analyses in which the effect of trait self-control would be statistically controlled for to see whether the willpower-mindset-moderation effect would still remain significant. Instead, we conducted a set of analyses to examine whether trait self-control would serve as a significant moderator of the ego-depletion effect. The results of these exploratory analyses (both fixed-effects and mixed-effects analyses) are reported in Tables F and G in S1 Appendix. The bottom line is that there was no evidence for such a moderation effect by trait self-control.

### Discussion

The current preregistered study failed to reproduce either the overall ego-depletion effect or its moderation by willpower mindset reported by Job et al. [1]. This was the case even though we used (a) a larger sample (*N* = 187) than the original study (*N* = 60) and (b) the same two depletion and outcomes tasks (the instructions and texts for the letter cancelation task were obtained from Job et al. [1]).

We should acknowledge again, however, that our study was likely underpowered (< .80 power) to reliably detect not only the main ego-depletion effect but also the hypothesized moderating influence of willpower mindset on ego depletion, especially for the planned fixed-effects analyses based on regressions. As noted earlier, when we initially designed and preregistered the study, there were no comparable studies that tested the impact of willpower mindset on ego depletion that we could have used to derive solid effect size estimates for the

moderation effect and to conduct a proper a priori power analysis. Thus, from the perspective of statistical power, this replication failure, in and by itself, does not provide a foolproof case against the replicability of the original Job et al. [1] study.

At the same time, we should also point out that, given what we now know about the likelihood of the ego-depletion effect itself [8, 11–13], conducting a sufficiently powered study to replicate Job et al.'s [1] original finding would have been extremely challenging, if not totally impossible. This is because, even with a somewhat generous effect-size estimate of $d = .10$ [13], it would have been necessary to have a sample of over 3,000 to detect a likely small (if any) ego-depletion effect with a power of .80, using an independent-sample $t$ test. Detecting a (non-crossover) moderation effect involving an individual-differences variable would have required an even larger sample size to achieve the same level of statistical power [14], although using proper mixed-effects modeling might help increase statistical power [33]. This means that, in terms of sample sizes, even the three existing multilab ego-depletion studies ($N$'s ranging from 1,775 to 3,531 [11–13]) might not be large enough to test Job et al.'s [1] moderation effect with statistical power of .80.

## Possible reasons for the current replication failure

Why might we have failed to replicate the original results reported by Job et al. [1]? As noted by Simonsohn [35], evaluating whether replication results are consistent with those of the original study is a complicated matter, but we discuss some possibilities here.

One possible reason is that we analyzed the data differently than Job et al. [1] did. We initially planned and tried to replicate Job et al.'s [1] analysis exactly, but, because the models implemented in R did not converge, we opted for conducting mixed-effects analyses instead without any covariates (e.g., age, trial numbers). Thus, we do not know what would have happened if we had successfully analyzed our data in the same way as Job et al. [1] did. It is therefore still possible (although we suspect unlikely) that not nesting trials under participants and/or including some covariates might have reproduced the pattern observed in the original study.

Conversely, it also seems possible that Job et al. [1] would not have observed a significant willpower-mindset moderation effect if their data had been analyzed with the methods reported here (fixed-effects or mixed-effects modeling, without including theoretically unmotivated covariates). If this were the case, our results might not be a case of replication failure at all. Because we were not able to obtain Job et al.'s [1] original dataset despite requests, we do not know what would have happened with such reanalysis.

Another possible reason for our replication failure is that, unlike Job et al. [1], we did not obtain a significant main effect of the ego-depletion effect: Perhaps the significant moderation effect of willpower mindset would be obtained only when the overall main effect of ego depletion itself is present. Although we cannot rule out this possibility, some of the recent large-scale studies—most notably the two studies presented in the Garrison et al. [36] article and Dang et al.'s [13] multilab replication study—reported a significant (albeit small) overall effect of ego depletion, but the evidence for the moderation effect was absent [36] or ambiguous [13]. Thus, even when the overall ego-depletion effect is significant, there is no guarantee that the moderating effect of ego depletion would always be obtained (see also [22] for a review).

Yet another possible reason for our replication failure is that, although similar in many respects, our study was not a direct replication of the original study and, hence, that some changes we introduced to improve the study design may have contributed to the discrepant results. Although some changes we introduced are unlikely to matter much (e.g., administering a second extension block for Stroop after the main replication block, reversing the

willpower-mindset scale in directionality to make it consistent with the trait self-control scale), we agree that other changes might. We point out two such changes here.

First, in our study, we administered various manipulation-check ratings (e.g., effort, difficulty) at multiple time points (see Fig 1). In contrast, Job et al.'s [1] original study did not include any such ratings at all (i.e., no manipulation-check results reported). Although some have argued that providing convincing manipulation-check results is critically important [37], it is also possible that we overdid it in our study and alerted participants to what this study was generally about, even though none of them were able to come up with the primary moderation hypothesis being tested in our study. This is a possibility that we cannot rule out without conducting further research.

Second, in our personal communications, Veronika Job suggested that a critical difference between our study and their study might be that we did not use a blurry text in the depletion condition for the letter cancelation task to avoid what we perceived to be an experimental confound (i.e., the depletion and control versions of this letter cancelation task differed not only in the hypothesized self-control demand but also in the visual appearance or blurriness of the text). According to Job's account, using an unprofessionally looking text might more likely make participants feel fine to "slack off" on the subsequent task, and this reduced motivation might magnify the impact of another motivation-related variable, willpower mindset.

This alternative explanation is interesting and worth testing, but this logic behind the use of a blurry text in the depletion condition is not specified in the original article. In fact, as noted earlier, Job et al. [1] stated that the main purpose of using the two blocks of the letter cancelation task was to "establish a behavioral pattern" in Block 1 and then to require participants in the depletion condition to "inhibit the previously established response" (p. 1687) in Block 2. Thus, the impact of using a blurry text on participants' motivation hypothesized by Job remains a speculative conjecture at this point, especially without conducting a further study that directly compares the use of blurry versus crisp texts in the depletion condition.

More generally, if the hypothesized moderation effect indeed hinges critically on the visual quality of the text used in the depletion condition, then such stimulus specificity of the effect would make it impossible for other researchers to test the hypothesized moderation effect using other depletion tasks and unambiguously predict whether they should expect a significant moderation effect in their study. Such theoretical vagueness is a major conceptual problem that we have argued before is highly prevalent in the ego-depletion literature [4], including not only the strength model [15] but also Job et al.'s [1] willpower-mindset account.

## Evaluating the replicability of Job et al.'s original finding

Although the reasons for our failure to replicate Job et al.'s [1] original finding are not clear, the results of our replication study are consistent with those of recent relevant studies. In fact, we recently conducted a qualitative review of the existing ego-depletion studies that tested the replicability of the hypothesized moderating influence of willpower mindset on ego depletion [22]. We found that, of the 13 qualifying studies (from 11 papers), 5 studies (including Job et al.'s [1] original study) reported some evidence for the moderation effect (mean $N = 464.5$; median $N = 96.5$), whereas 8 studies failed to observe this effect (mean $N = 834.3$; median $N = 275.5$), including the current study [12, 36, 38–41].

Importantly, among the 5 studies that provided some evidence for the moderation effect [1, 13, 42, 43], only one study by Chow et al. [42] provided clean statistical evidence for the original finding. The evidence reported in the other four studies provided ambiguous support, starting with the original Job et al. [1] study (not properly nesting trials under participants and including unusual covariates). Dang et al.'s [13] multilab replication study also demonstrated

some evidence of the moderation effect, but the effect was restricted to the analysis involving a secondary dependent measure (RTs for the antisaccade task); the moderation effect was non-significant for the analysis involving the primary dependent measure (accuracy). Even this moderation involving the secondary RT measure was tenuous in that the effect became non-significant when the analyses excluded participants who hence performed at chance level on the antisaccade task.

Finally, two studies conducted by Singh and Göritz [43] (Studies 1 & 2), both using the same combination of the depletion and outcome tasks as Job et al.'s [1] study and our study, also observed a moderating influence of willpower mindset in their data (especially in Study 1). Strikingly, however, the moderation pattern was opposite of what Job et al. [1] observed: a *greater* ego-depletion effect for individuals holding a *nonlimited*-willpower, rather than limited-willpower, mindset. Thus, even though these two studies reported some moderating influence, the results contradict Job et al.'s [1] unidirectional account (i.e., limited mindset → larger depletion).

Our review [22] also revealed that the 5 studies that provided some evidence for the moderation effect differed from the 8 studied that failed to find such evidence in some systematic ways. Although we will not elaborate here (see [22] for details), the most striking difference between the two categories of studies concerns the status of preregistration and data sharing: Nearly all (7 out of 8) studies that failed to observe the moderation effect are preregistered and have shared the data, but that was not the case for most (4 out of 5) studies that found some evidence for the moderation effect (the sole exception was Dang et al.'s [13] aforementioned multilab study). This means that Job et al.'s [1] original finding has not been unambiguously replicated yet in a large-sample preregistered study with publicly shared data.

In summary, although the statistical power of any individual studies (including ours) may not have achieved high statistical power of .80, the existing evidence for the hypothesized moderating influence of willpower mindset on the ego-depletion effect is weak at best. Although Job et al.'s [1] finding is well known and has been influential, it is not a replicable finding that can provide a solid basis for rigorous theoretical development.

## Limitations and qualifications

Although we already pointed them out explicitly in various places, we would like to summarize the limitations of our study in a single subsection. First, despite using a larger sample ($N = 187$) than in the original study ($N = 60$), the study was still underpowered to detect the hypothesized moderation effect (especially in our fixed-effects modeling). Second, our final protocol deviated from the preregistered version in several ways, although such protocol deviations were all minor. Third, although our replication study had many similarities to the original Job et al. [1] study, it was not a direct replication in that we introduced a host of changes to improve the design (e.g., administering manipulation-check ratings at multiple time points, not using blurry texts in the depletion condition). Thus, any of the changes we introduced may have led to the failure to replicate the original study. Finally, because we were not able to analyze our data and Job et al.'s [1] data in the same ways, we cannot evaluate the extent to which the hypothesized moderation effect critically hinged on the unconventional data-analytic method used in the original study. These are all important limitations that one should keep in mind in interpreting our results.

Although we critiqued Job et al.'s [1] study, we would like to emphasize here that our critique is restricted to their specific application of willpower mindset to the controversial phenomenon of ego depletion. In fact, we believe that the concept of willpower mindset itself is highly useful in self-control research (see [44] for a review), just like that of growth/fixed

mindset of intelligence has long been in education research [2]. Thus, even though we are not convinced of the replicability of Job et al.'s [1] original moderation finding and claim based on the ego-depletion paradigm, that does not mean that other non-ego-depletion research findings based on the notion of willpower mindset [44] are also problematic.

## Concluding remark

As the current study and other recent studies reviewed above have shown, the influential moderation effect initially reported by Job et al. [1] may not be as robust as has been portrayed in the scientific literature as well as in the popular media [16–18]. Perhaps this lack of replicability may not be a surprise, considering that the replicability of the ego-depletion effect itself has been seriously questioned lately. Far from being the "biggest bombshell" evidence [16], Job et al.'s [1] original finding should be interpreted with great caution, unless a future preregistered replication study can provide compelling and unambiguous evidence for the hypothesized moderation effect on ego depletion.

Given that doing so with sufficient statistical power requires a really large sample size, an obvious question is whether undertaking such a major enterprise is warranted, especially in light of the scant evidence that currently exists for the replicability of Job et al.'s [1] original finding [22]. Our answer is a resounding "no," at least until more effective ways of testing ego depletion in everyday, real-life settings are developed, where the impact of motivation and related factors (e.g., mental effort, willpower mindset) is likely much more substantial and meaningful. Trying to further replicate the ego-depletion effect and its moderation by willpower mindset in artificial laboratory settings—especially using short, contrived tasks (such as the letter-cancelation task) that have not even been validated as effective measures of self-control [4]—may not be the best use of anyone's time and effort.

## Supporting information

**S1 Appendix. Additional analyses.**
(PDF)

## Acknowledgments

We thank Veronika Job for providing the stimulus materials and discussing with us the analyses and results of our respective studies; Clayton Ickes, Julia Squeri, Isabella Conte, Marjorie McIntyre, and Joellen Fresia for data collection; Michael Kane and John Lurquin for their input on earlier drafts; and Michael Inzlicht, Aaron L. Wichman, and an anonymous reviewer for their helpful feedback on an earlier version of this manuscript.

## Author Contributions

**Conceptualization:** Nicholas P. Carruth, Akira Miyake.

**Data curation:** Nicholas P. Carruth, Jairo A. Ramos.

**Formal analysis:** Nicholas P. Carruth, Jairo A. Ramos.

**Investigation:** Akira Miyake.

**Methodology:** Nicholas P. Carruth, Akira Miyake.

**Project administration:** Nicholas P. Carruth, Akira Miyake.

**Resources:** Akira Miyake.

**Software:** Akira Miyake.

**Supervision:** Akira Miyake.

**Visualization:** Nicholas P. Carruth, Jairo A. Ramos.

**Writing – original draft:** Nicholas P. Carruth, Akira Miyake.

**Writing – review & editing:** Nicholas P. Carruth, Jairo A. Ramos, Akira Miyake.

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
