## [Decision Letter · Decision Letter 0]

22 May 2023

PONE-D-23-10991Does Willpower Mindset Really Moderate the Ego-Depletion Effect? A Preregistered Replication of Job, Dweck, and Walton (2010)PLOS ONE

Dear Dr. Miyake,

Thank you for submitting your manuscript to PLOS ONE. After careful consideration, we feel that it has merit but does not fully meet PLOS ONE’s publication criteria as it currently stands. Therefore, we invite you to submit a revised version of the manuscript that addresses the points raised during the review process.

We look forward to receiving your revised manuscript.

Kind regards,

Dr Stergios Makris

Academic Editor

PLOS ONE

Journal Requirements:

Reviewers' comments:

Reviewer's Responses to Questions

**Comments to the Author**

1. Is the manuscript technically sound, and do the data support the conclusions?

Reviewer #1: Yes

Reviewer #2: Yes

Reviewer #3: Yes

2. Has the statistical analysis been performed appropriately and rigorously? 

Reviewer #1: Yes

Reviewer #2: Yes

Reviewer #3: Yes

3. Have the authors made all data underlying the findings in their manuscript fully available?

Reviewer #1: Yes

Reviewer #2: Yes

Reviewer #3: Yes

4. Is the manuscript presented in an intelligible fashion and written in standard English?

Reviewer #1: Yes

Reviewer #2: Yes

Reviewer #3: Yes

5. Review Comments to the Author

Reviewer #1: This is a nice paper, and I am in favour of it being published. It shows, yet again, that past studies of ego depletion cannot be replicated. That is, a close(ish) replication could not demonstrate the “classic” ego depletion effect. What is more, participants’ mindsets about the nature of willpower do not moderate any depletion effect, in contrast to an influential study by Job et al.

Like I said, I think this should be published. But I have some comments, nonetheless. First, why even bother with further replications of this now more-or-less dead research area? Except for a very small number of partisans, no one believes ego depletion (at least as produced in lab settings) is real/replicable. And if ego depletion is not real, there is nothing to moderate, including willpower mindsets. The field has moved on. So, no one will be surprised by these results. That said, I still think all such evidence should be published even if these are not novel results.

Second, the authors call their replication a direct/close replication and while I think it is close, there are enough deviations from the original study that I am no longer sure how close this replication is. Replications vary on a continuum of closeness, with exact on one end and conceptual on the other. I don’t think the current replication should be considered conceptual, but nor do I think it should be considered exact or even close. It is relatively close, but not close. Do I think anything would be different if the authors did an exact/close replication? No, absolutely not. I do not think any of this would replicate; nor do I think scholars should devote more time and money to finding out. Like I said, this is a more or less settled area. But I would advise the authors to do even more to make clear that this was a close-ish replication, and not exact. They should be even more open to the possibility that their various deviations from the original (and there were quite a few of them; not just the bluriness of the letter-cancelation task) were consequential.

Third, I was not persuaded by the sample size justification. The authors suggest that they had no real idea of the effect size and seemed to use an n=100 per cell rule of thumb. With this sample size, the authors suggested they had 80% power to detect an effect size of over d=.40. But given all we know now, and even what we knew back in 2016-2017 when the study was run, no one should expect such a large effect size. The effect size, to the extent that it is different from zero, is likely d=.10. But you would need over two thousand participants to get 80% power to detect such an effect. With a more reasonable to expect d=.20, you would need over 600 participants. Like I said above, I don’t think anyone should bother trying to replicate ego depletion anymore, so I am not suggesting that the authors run more participants. This would be a waste of time. However, that this replication is underpowered (even if significantly better powered than Job et al) is a major limitation, and the authors should be forthright in listing and discussing this major limitation (i.e., albeit very unlikely, they could be experiencing Type II error).

I hope my comments are helpful to the author and editor.

I sign all my reviews,

Michael Inzlicht

Reviewer #2: The authors conducted a close-to-direct replication of Job, Dweck, and Walton's study examining the moderating role of willpower beliefs on self-control depletion. The authors improved on the design and analysis of the original study in numerous ways. Given the rigorous nature of the design and analysis, the results do indeed a quite compelling argument against Job and colleagues findings. The theoretical contribution of the current manuscript to our understanding of the nature of ego-depletion (if any) is notable. The current manuscript should be accepted pending minor revisions.

Some suggested revisions: 1) It is not always clear why certain liberties were taken to not follow the preregistration plan exactly. For instance, lines 201-203 just states that they had planned to administer a questionnaire but it ultimately wasn't administered. No explanation is given for why it wasn't. Better elucidating this and others would be helpful. 2) The paragraph comprising lines 432 - 439 details a MANOVA that was conducted on the combination of effortful, tired, difficult, boring, frustrated, and effort. It would be helpful to show the correlations between these rating as a justification for why they were combined in a MANOVA. 3) A recommendation for a future study examining this question. The current study contains numerous motivation and manipulation check questions scattered throughout the study. Administering so many of these during the study could be introducing extra "noise" making it harder to uncover a significant effect. It may be worthwhile replicating the study without all of these to see if it cleans up the variance (although I understand why the authors included them in their current study).

Overall, I recommend accepting pending minor revisions.

Reviewer #3: Thank you for an excellent article. Your introduction set the stage well for this contribution, and you analytic work was thorough. I have two comments. The first, non-substantive, comment is that the language describing the sample size in the abstract was hard for me to understand. Specifically, I got the initial impression that your replication might have had an N = 60, which is not what you did.

My second comment is potentially challenging. You have powered the study to detect an effect of d ~.40, but one might argue that our optimistic best guess effect size for ego-depletion is d ~ .10. If the effect size is actually .10 (Dang et al., 2021), this study could have missed this effect, meaning this non-replication is actually the case of an under-powered study.

I know there are Bayesian approaches to arguing for a null effect, but I know nothing about them. Something that could be useful from a Frequentist perspective is Lakens (2017) in SPPS.

I would feel much more comfortable with this paper if the authors could make a stronger argument for the null hypothesis. However, I defer to our editor for the best solution to this problem.

Thank you for a well-done paper.

6. PLOS authors have the option to publish the peer review history of their article (what does this mean?). If published, this will include your full peer review and any attached files.

Reviewer #1: **Yes: **Michael Inzlicht

Reviewer #2: No

Reviewer #3: **Yes: **Aaron L. Wichman

---

## [Author Response · Author response to Decision Letter 0]

27 May 2023

We greatly appreciate thoughtful and timely feedback from the three reviewers on our replication study of Job, Dweck, & Walton (2010). On the basis of the reviewers’ comments, we thoroughly revised our manuscript. 

Because we carefully went through the entire manuscript and made many minor edits to improve the exposition (in addition to the more substantial ones responding to the reviewers’ comments), the version with full-on track changes is really difficult to read. So we decided to submit (a) a clean version without any markings (named “Manuscript”) and (b) a version in which more substantial changes are indicated in light (named “Revised Manuscript with Track Changes”) via Editorial Manager, as per your instructions. 

Below, we provide our point-by-point responses to the individual reviewer comments. Most of the substantial changes occurred in the Discussion section.

===

Reviewer #1

Reviewer Comment: This is a nice paper, and I am in favour of it being published. It shows, yet again, that past studies of ego depletion cannot be replicated. That is, a close(ish) replication could not demonstrate the “classic” ego depletion effect. What is more, participants’ mindsets about the nature of willpower do not moderate any depletion effect, in contrast to an influential study by Job et al.

Our Response: We thank Reviewer 1 for his generally positive and very thoughtful feedback.

Reviewer Comment: Like I said, I think this should be published. But I have some comments, nonetheless. First, why even bother with further replications of this now more-or-less dead research area? Except for a very small number of partisans, no one believes ego depletion (at least as produced in lab settings) is real/replicable. And if ego depletion is not real, there is nothing to moderate, including willpower mindsets. The field has moved on. So, no one will be surprised by these results. That said, I still think all such evidence should be published even if these are not novel results.

Our Response: As Reviewer 1 is clearly aware, we conducted this study a long time ago, when our knowledge of the replicability of the ego-depletion effect, let alone the replicability of Job et al.’s moderation finding, was more limited (our designing of this study had already started when Hagger et al.’s influential first multilab replication study was published). Because it took us such a long time to submit this manuscript for publication, we fully understand Reviewer 1’s “why even bother further replications?” question. We would not conduct a new large-scale replication study now on this topic.

We decided to submit this paper to PLOS ONE because we believe that this is still good-quality work (the three reviewers all seem to agree!), and we also did not want to contribute to the so-called “file drawer” problem by not publishing a replication failure (critically important for future meta-analyses). Moreover, regardless of the general doubts surrounding the ego-depletion phenomenon, Job et al.’s (2010) work continues to be popular and frequently cited. So, even though the current replication failure may not be novel now, we believe that it is still worth publishing this work (as noted by Reviewer 1). We chose PLOS ONE for this work because the journal does not focus on “novelty” as a publication criterion.

Reviewer Comment: Second, the authors call their replication a direct/close replication and while I think it is close, there are enough deviations from the original study that I am no longer sure how close this replication is. Replications vary on a continuum of closeness, with exact on one end and conceptual on the other. I don’t think the current replication should be considered conceptual, but nor do I think it should be considered exact or even close. It is relatively close, but not close. Do I think anything would be different if the authors did an exact/close replication? No, absolutely not. I do not think any of this would replicate; nor do I think scholars should devote more time and money to finding out. Like I said, this is a more or less settled area. But I would advise the authors to do even more to make clear that this was a close-ish replication, and not exact. They should be even more open to the possibility that their various deviations from the original (and there were quite a few of them; not just the blurriness of the letter-cancelation task) were consequential.

Our Response: We believe that this replication of Job et al.’s moderation effect is the closest one to the original study, but we accept Reviewer 1’s point that we made a host of changes to the original protocol used by Job et al. We justify why we made these changes, but Reviewer 1 is right in that some of the changes we intentionally made to improve the study design might have affected how participants would perform the task and hence the outcome of the study results. 

For this reason, we carefully went through the entire manuscript and removed any suggestions that our study is a close (or close-to-direct) replication of Job et al.’s original study (we resisted the temptation to use the phrase, “close-ish,” in our manuscript, even though it accurately describe our replication!). This means removing/rewriting some sentences from a few places and providing even more frank and explicit acknowledgment of the possibility that some changes we introduced to our procedure might have affected the results (see, for example, p. 33), as per Reviewer 1’s advice.

Reviewer Comment: Third, I was not persuaded by the sample size justification. The authors suggest that they had no real idea of the effect size and seemed to use an n=100 per cell rule of thumb. With this sample size, the authors suggested they had 80% power to detect an effect size of over d=.40. But given all we know now, and even what we knew back in 2016-2017 when the study was run, no one should expect such a large effect size. The effect size, to the extent that it is different from zero, is likely d=.10. But you would need over two thousand participants to get 80% power to detect such an effect. With a more reasonable to expect d=.20, you would need over 600 participants. Like I said above, I don’t think anyone should bother trying to replicate ego depletion anymore, so I am not suggesting that the authors run more participants. This would be a waste of time. 

However, that this replication is underpowered (even if significantly better powered than Job et al) is a major limitation, and the authors should be forthright in listing and discussing this major limitation (i.e., albeit very unlikely, they could be experiencing Type II error).

Our Response: Our sample size justification (at least what we said in our preregistration document) was based on what we knew back in 2016. Given what we now know about the likely magnitudes of the ego-depletion effect (if it indeed exists), we completely understand that our sample-size justification might not be compelling now (Reviewer 3 also raises this point). 

Because we do not want to engage in post-hoc rationalization of our sample sizes, we agree with Reviewer 1 in that the best course of action is to explicitly acknowledge that, in light of what we now know about the likely magnitudes of the ego-depletion effect, this study was underpowered to detect the hypothesized moderation effect. This acknowledgement was provided in multiple places in the manuscript, starting with p. 9 (when we report post-hoc power analysis results). We also acknowledge this power limitation issue on p. 31, although we also point out that many other studies, even those recent multilab replication studies, were likely underpowered.

More generally, to be more forthcoming about the limitations of our replication study, we created a subsection called “Limitations and Qualifications” (pp. 36-37), where we summarize the major limitations of the study mentioned in different parts of the manuscript. 

Finally, we would like to point out that we added a new paragraph to the very end of our paper to address a running theme in Reviewer 1’s comments about the current status of the ego-depletion literature. We fully agree that the ego-depletion effect, at least as has been studied in laboratory settings (e.g., Job et al.’s study), is not a robust, replicable phenomenon that is worth spending a lot of time and effort on anymore. We were hesitant to add such a paragraph at the very end at the time of initial submission to PLOS ONE, but Reviewer 1’s repeated remarks in his review motivated us to conclude our paper by essentially saying that it is not worth further replicating Job et al.’s moderation influence in light of what we now know.

Reviewer Comment: I hope my comments are helpful to the author and editor.

Our Response: We thank Reviewer 1 again for his helpful comments.

===

Reviewer #2 

Reviewer Comment: The authors conducted a close-to-direct replication of Job, Dweck, and Walton's study examining the moderating role of willpower beliefs on self-control depletion. The authors improved on the design and analysis of the original study in numerous ways. Given the rigorous nature of the design and analysis, the results do indeed a quite compelling argument against Job and colleagues findings. The theoretical contribution of the current manuscript to our understanding of the nature of ego-depletion (if any) is notable. The current manuscript should be accepted pending minor revisions.

Our Response: We thank Reviewer 2 for positive comments on our manuscript.

Reviewer Comment: Some suggested revisions: 1) It is not always clear why certain liberties were taken to not follow the preregistration plan exactly. For instance, lines 201-203 just states that they had planned to administer a questionnaire but it ultimately wasn't administered. No explanation is given for why it wasn't. Better elucidating this and others would be helpful. 

Our Response: Most of the minor procedural deviations from our preregistered protocol occurred because we were relatively new to the business of preregistration back then. Essentially, we completed the preregistration a bit too prematurely (before we completely finalized the study protocol) and failed to update the protocol appropriately before or right after the “real” data collection started. This point is now openly acknowledged on p. 7. We also added a bit more explanation of why such minor deviations from the preregistered protocol occurred where appropriate.

Reviewer Comment: 2) The paragraph comprising lines 432 - 439 details a MANOVA that was conducted on the combination of effortful, tired, difficult, boring, frustrated, and effort. It would be helpful to show the correlations between these rating as a justification for why they were combined in a MANOVA. 

Our Response: In response to this issue, we added a sentence about the correlations among difference ratings at the end of the section mentioning our exploratory MANOVA results (the bottom of p. 21). Basically, five of the six ratings were generally correlated with one another, but one of them (boring) was not, a pattern consistent with what we reported in the manipulation-check section. This is a minor issue in our view, so we decided against conducting the MANOVA again without the boring rating (especially given that, even including “boring” in the MANOVA, the manipulation-check results were consistent with our expectations).

Reviewer Comment: 3) A recommendation for a future study examining this question. The current study contains numerous motivation and manipulation check questions scattered throughout the study. Administering so many of these during the study could be introducing extra "noise" making it harder to uncover a significant effect. It may be worthwhile replicating the study without all of these to see if it cleans up the variance (although I understand why the authors included them in their current study).

Our Response: We agree that we administered manipulation-check ratings too frequently in our study and that doing so might have affected the overall results. Thus, as we noted above in our response to Reviewer 1, we more explicitly acknowledged this possibility on p. 33. However, we chose not to point out a need for conducting this future study suggested by Reviewer 2 to address this frequent ratings issue, because, as we noted in our reply to Reviewer 1, we just do not think that it is worth spending a lot of time and energy conducting ego-depletion experiments to address issues like this. In fact, for this reason, we also deleted a couple of sentences we had in the previous version in which we suggested conducting some future studies to test some specific ideas (e.g., comparing blurry vs. crisp texts for the letter-cancelation task). In light of what we know, it is simply not worth conducting such studies any more, and I hope Reviewer 2 would agree.

Reviewer Comment: Overall, I recommend accepting pending minor revisions.

Our Response: We thank Reviewer 2 again for their useful feedback.

===

Reviewer #3

Reviewer Comment: Thank you for an excellent article. Your introduction set the stage well for this contribution, and you analytic work was thorough. I have two comments. The first, non-substantive, comment is that the language describing the sample size in the abstract was hard for me to understand. Specifically, I got the initial impression that your replication might have had an N = 60, which is not what you did.

Our Response: We thank Reviewer 3 for his positive evaluation of our manuscript. We carefully revised the manuscript so that it was clear that the sample size of 60 was applicable to the original study by Job et al., not to our study.

Reviewer Comment: My second comment is potentially challenging. You have powered the study to detect an effect of d ~.40, but one might argue that our optimistic best guess effect size for ego-depletion is d ~ .10. If the effect size is actually .10 (Dang et al., 2021), this study could have missed this effect, meaning this non-replication is actually the case of an under-powered study. I know there are Bayesian approaches to arguing for a null effect, but I know nothing about them. Something that could be useful from a Frequentist perspective is Lakens (2017) in SPPS. I would feel much more comfortable with this paper if the authors could make a stronger argument for the null hypothesis. However, I defer to our editor for the best solution to this problem.

Our Response: We thank Reviewer 3 to raise this power issue also raised by Reviewer 1. As noted above, we addressed this issue by primarily acknowledging that our study (as well as many other existing studies) was underpowered in light of what we now know about the ego-depletion effect.

We appreciate Reviewer 3’s suggestions for possible options for addressing this power issue (e.g., Lakens’ equivalence tests, Bayesian analyses). In our earlier ego-depletion work (Lurquin et al., 2016, published in PLOS ONE), we reported Bayes factors to supplement our null significance test results, but, in this particular case, the main focus is on the statistical moderation effect (an interaction between an experimental manipulation and an individual-differences variable) in the context of regression models or mixed-effects modeling. Although equivalence tests and Bayes factors are more easily applicable to the tests of the ego-depletion effect (i.e., comparing two means), we are not statistically savvy enough to apply these methods to the tests of moderation effects. Given that most (if not all) of existing studies (beyond our study) were likely underpowered to detect the hypothesized moderation effects, we hope that our frank acknowledgment of the underpowered nature of our study to detect the moderation effect would be sufficient. 

Instead of conducting additional power-related analyses, however, we strengthened our discussion of the existing ego-depletion studies that reported some evidence on the hypothesized moderation effects (see pp. 34-36), primarily by briefly summarizing the key findings from our now-in-press brief review paper (Miyake & Carruth, in press). Although our description in this manuscript had to be concise, we make it clear that the existing evidence for the hypothesized moderation effect is weak at best (i.e., we were able to find only one existing study that provided unambiguous evidence for the hypothesized moderation effect by willpower mindset). We hope that providing such a more thorough consideration of the existing literature may help Reviewer 3 feel more at ease with the lack of strong sample size justifications for our study.

Reviewer Comment: Thank you for a well-done paper.

Our Response: We thank Reviewer 3 again for his feedback.

===

Additional Issues

In addition to the point-by-point responses provided above, we would like to mention a few other things about our revision.

First, as we noted earlier, most of the substantial revisions occurred in the Discussion section. This is why most parts of that section are highlighted in light blue. We now have subsection headers for the Discussion section, and various additions are made to fully address the points raised by the reviewers. As already mentioned above, for example, we added (a) a new concluding paragraph, (b) a new limitations/qualifications section (in which we now also explicitly point out that our critique of the willpower mindset concept is applicable only to its application to the ego-depletion literature), and (c) an expanded discussion of the existing literature on the replicability of Job et al.’s original moderation finding.

Second, because we expanded our discussions where appropriate (especially in the Discussion section), we had to add additional references. To be transparent about the added references, we highlight them in green in the Tracked Changes file (we use green in the references section because blue is used for URL links in the reference list). In a few places, we added some references to provide further justifications for our deviations from Job et al.’s original protocol.

Finally, although we already mentioned this above, we have a recently accepted paper at Frontiers in Psychology in which we conduct a qualitative review of all the published studies that reported some evidence on the moderation of the ego-depletion effect by willpower mindset (Miyake & Carruth, in press). Because it is a review/opinion paper and this PLOS ONE manuscript is primarily about an empirical study (preregistered replication), we believe that they are distinct. We also tried to avoid too much overlap when describing the key findings of our review when revising the current manuscript. Because this brief review was officially accepted by Frontiers just a few days ago (May 24), so the final formatting version is not available yet, but a preliminary version can be viewed at the following URL:

https://www.frontiersin.org/articles/10.3389/fpsyg.2023.1208299/full

We thank again the editor and the three reviewers for the prompt and careful review/editorial process. We hope that our revisions were satisfactory and that our paper is now ready for publication in PLOS ONE.

---

## [Editor Report · Decision Letter 1]

15 Jun 2023

Does willpower mindset really moderate the ego-depletion effect? A preregistered replication of Job, Dweck, and Walton (2010)

PONE-D-23-10991R1

Dear Dr. Miyake,

We’re pleased to inform you that your manuscript has been judged scientifically suitable for publication and will be formally accepted for publication once it meets all outstanding technical requirements.

Kind regards,

Stergios Makris

Academic Editor

PLOS ONE
---

## [Editor Report · Acceptance letter]

22 Jun 2023

PONE-D-23-10991R1 

Does willpower mindset really moderate the ego-depletion effect? A preregistered replication of Job, Dweck, and Walton (2010) 

Dear Dr. Miyake:

I'm pleased to inform you that your manuscript has been deemed suitable for publication in PLOS ONE. Congratulations! Your manuscript is now with our production department. 

Kind regards, 

on behalf of

Dr. Stergios Makris 

Academic Editor

PLOS ONE